# MicroRNA-Assisted Hormone Cell Signaling in Colorectal Cancer Resistance

**DOI:** 10.3390/cells10010039

**Published:** 2020-12-30

**Authors:** Crescenzo Massaro, Elham Safadeh, Giulia Sgueglia, Hendrik G. Stunnenberg, Lucia Altucci, Carmela Dell’Aversana

**Affiliations:** 1Department of Precision Medicine, University of Campania “Luigi Vanvitelli”, Via De Crecchio, 7, 80138 Naples, Italy; crescenzo.massaro@unicampania.it (C.M.); elham.safadeh@unicampania.it (E.S.); giulia.gueglia@unicampania.it (G.S.); 2Prinses Maxima Centrum, Heidelberglaan 25, 3584 CS Utrecht, The Netherlands; H.G.Stunnenberg@prinsesmaximacentrum.nl; 3Institute of Experimental Endocrinology and Oncology “Gaetano Salvatore” (IEOS)-National Research Council (CNR), Via Sergio Pansini 5, 80131 Naples, Italy

**Keywords:** microRNA, hormone signaling, colorectal cancer, drug resistance

## Abstract

Despite substantial progress in cancer therapy, colorectal cancer (CRC) is still the third leading cause of cancer death worldwide, mainly due to the acquisition of resistance and disease recurrence in patients. Growing evidence indicates that deregulation of hormone signaling pathways and their cross-talk with other signaling cascades inside CRC cells may have an impact on therapy resistance. MicroRNAs (miRNAs) are small conserved non-coding RNAs thatfunction as negative regulators in many gene expression processes. Key studies have identified miRNA alterations in cancer progression and drug resistance. In this review, we provide a comprehensive overview and assessment of miRNAs role in hormone signaling pathways in CRC drug resistance and their potential as future targets for overcoming resistance to treatment.

## 1. Drug Resistance in Colorectal Cancer

Colorectal cancer (CRC) is the third cause of cancer-related death worldwide after lung and breast carcinomas [1]. Current strategies for the treatment of CRC include surgery, radiotherapy, targeted therapy, and chemotherapy [2]. Despite the availability of various approaches, therapeutic efficacy and the current estimation of survival for treated patients are low, indeed, almost 50% of surgically or non-surgicallymedically treated patients experience relapse [3]. The problem of ineffectiveness of therapy/drug resistance could be due to the considerable heterogeneity of CRC, which can be categorized into three main groups: (1) Chromosomal Instability (CIN) with ~65–85% of incidence, (2) Microsatellite Instability (MSI) present in ~12–15% of patients, and (3) CpG Island Methylator phenotype (CIMP) in ~10–20% of patients [4].CIN is a genomic instability in which part of the chromosome (or the chromosome itself) can be duplicated or erased, leading to a series of abnormalities. Among these, aneuploidy represents the main modification in CRC [4]. This chromosomal aberration is not only associated with poor prognosis but also with the acquisition of intrinsic resistance to drugs [5], in particular, resistance to irinotecan is indeed partially associated with the aneuploidy state of HCT116 and LoVo cells [6]. Further, loss of 19p13.3 and/or 16p13.3 are associated with reduced Overall Survival (OS) in CRC patients as they comprise apoptosis-related genes [7]. In this regard, 16p13.3 region includes *AXIN1* gene whose expression induces apoptosis in CRC [7], whereas 19p13.3 region carries the pro-apoptotic gene *BAX* resulting in 5-FU resistance in CRC. Chromosomal segment deletions also concern loss of epigenetic regulatory factors whose lack has been associated with 5-FU resistance [8]. The accumulation of molecular anomalies is one of the major causes of CRC formation and progression, especially regarding tumor suppressor genes. The most widespread mutation in early-stage CRC involves the adenomatous polyposis coli (APC) gene, a negative regulator of the WNT signaling pathway. Dysregulation or absence of APC leads to the accumulation and translocation of β-catenin in the nucleus, which ultimately is associated with the progression of CRC [9]. Characterization of truncating mutations in APC was performed in 58.2% of primary CRCs, also correlated with resistance to 5-fluorouracil (5-FU) by alteration of the Wnt/β-catenin signaling cascade [10]. Other gene mutations commonly implicated in these CRC-dysregulated signaling pathways are PI3K [11], KRAS and BRAF. Furthermore, compared to wild-type *KRAS*, mutated *KRAS* is rarely associated with MSI in CRC patients, whereas *BRAF* mutations, which occur in about 4–20% of CRCs, are linked to MSI in colorectal tumorigenesis [12]. MSI are short-length repetitive DNA sequences thatare involved in cellular DNA repair mechanisms and their mutations, in particular frameshift and base substitution mutations are associated with Lynch syndrome and the onset of CRC [13]. CRC patients showing MSI can be divided into high level (MSI-H) and low level of MSI (MSI-L) groups [14], where MSI-H wasfound in 10–15% of cases [15]. The relationship between MSI related sites and tumor drug resistance has been demonstrated, mostly to cisplatin [16,17] and 5-fluorouracil [18] treatments. MSI is associated with mismatch repair deficiency (MMR-D), which arises from the loss of function of mismatch repair (MMR) genes [19]. Such aberrant expressions are harnessed by cancer also for immunotherapy escape: Indeed, MSI-H/MMR-D CRC patients showed multiple genetic alterations related to immune response, in particular in WNT/β-catenin signaling [20], whose activation reduces the efficacy of immune checkpoint blockade therapy [20]. Furthermore, resistance may also develop from dysregulation of nucleotide excision repair (NER) pathway and the expression levels of DNA repair proteins, such as ERCC1, XRCC1, and XDP [21], in particular, regarding platinum-based drugs, such as oxaliplatin. Structurally, oxaliplatin carries a 1,2-diaminocyclohexanelig and platinum groups that induce DNA damage in a way thatis more complex for cells to repair [21], but an increased amount of DNA repair proteins helpsto overcome such molecular injury.

Besides genetic modifications, also aberrant DNA methylation is a cancer hallmark consisting of DNA hypo/hyper-methylation, which modulates gene expression. The CIMP consists of significant hypermethylation of the CpG islands of tumor suppressor genes, which leads to their inactivation and thus promotes tumor progression. Epigenetic alterations are associated with resistance to commonly used drugs in CRC, such as 5-fluorouracil (5-FU), oxaliplatin, and irinotecan [22]. Actually, most of the aberrant expression of various genes is due to epigenetic modifications rather than a genomic modification, thus making epigenetics noteworthy in CRC to assess in detail [23]. For instance, CRC cells in which the cell cycle regulator p16 is methylated show considerable resistance to irinotecan [24], as well as the hypermethylation of particular genes, such as SAT, C8orf4, LAMB3, is responsible for the decrease in sensitivity to cisplatin [25]. Moreover, the use of known epigenetic drugs, such as HDAC inhibitors, exhibited a re-sensitization of CRC cells to common drugs, such as 5-fluorouracil [26]. Moreover, the demethylating agent 2′-deoxy-5-azacitidine (DAC), an inhibitor of DNMT, allowed the reactivation of MLH1, whose repression is associated with resistance to various drugs, including cisplatin and epirubicin. This reactivation has, therefore, granted an increase in the effectiveness of chemotherapy in CRC cells [27]. In addition, miRNAs showed to underpin a prominent role in conferring drug resistance in CRC. As a matter of fact, miR-34a, downregulated in CRC, was shown to heighten 5-FU sensitivity though restraining SIRT1 expression [28,29,30]. High levels of SIRT1 were also associated with diminished expression of miR-29b [31], leading to oxaliplatin-resistance. miR-140 was found increased in CRC cells, thereby, inhibiting the expression of HDAC4, which contributed to G1 and G2 phases arrest, and finally resulting in 5-FU resistance in CRC cells [32]. Cancer cells can also embrace drug resistance by modulating intracellular drug concentration or its metabolism. The efflux of anti-tumor drugs limits the intracellular drug accumulation, which leads to chemotherapy failure. These processes are mediated by different membrane transporters that regulate the balance between drug uptake and efflux, such as the ATP-binding cassette (ABC) family [33]. Several ABC transporters are upregulated in CRC, causing a reduction in therapeutic efficacy [33]. P-glycoprotein (P-gp) is one of the 48 identified human ABC transporters overexpressed in myeloma, leukemia, and gastrointestinal cancers. High expression of P-gp was observed at CRC diagnosis, suggesting the existence of an intrinsic resistance mechanism [33]. Furthermore, several other ABC transporters, such as MRP1and BCRP, can be induced by anticancer drugs during colorectal tumorigenesis [33]. Overexpression of ABC transporters is one of the main mechanisms that gastrointestinal carcinomas exploit toward cisplatin resistance, enforcing the expression of a typical drug-resistance phenotype [34]. Furthermore, such transporters are significantly expressed in Cancer Stem Cells (CSCs) [21], which seems to be an important chapter in drug resistance [35]. The developed resistance may further be related to changes in the expression of enzymes responsible for drug metabolism, such as Thymidine Phosphorylase (TP). TP enzyme converts 5-FU toward fluorodeoxyuridine (FUDR,) one of its active forms [36], aberrant expression of TP is associated with 5-FU resistance [37]. On the other hand, drug responsiveness may associate with the expression of enzymes targeted by drugs, such as Thymidylate Synthase (TS) in 5-FU treatment [38,39,40]. Patients with low TS expression respond positively to 5-FU therapy, whereas patients with high TS levels do not respond to therapy and have a worse OS [41,42]. Figure 1 illustrates the discussed mechanisms exploited by CRC cells to overcome antitumoral treatments.

## 2. The Role of Hormone Signaling in CRC Resistance

Hormones are released from endocrine glands into the blood to regulate different stimuli in receiving cells, thus modulating their metabolism, growth, and development [43]. In receiving cells, hormone signaling comprises hormone binding to its receptor and signal transduction through different functional pathways. Several hormone signaling pathways are altered in CRC, often resulting in drug resistance. Tumor cells can avoid drug-mediated apoptosis at different levels of signaling pathways as these typically have multiple components. Most hormone signaling pathways exert their function through PI3K/Akt cascade: Upon hormone stimulation, PI3K is phosphorylated and its activity is mediated by RasandSrcfamilykinases that phosphorylate Akt, which, in turn, regulates the activation of proteins involved in apoptosis, metabolism, protein synthesis, and cell division [44]. Furthermore, the 3R3 subunit of PI3K is upregulated in CRC and controls the expression of TP via the NF-kB pathway, thus preventing the activation of drugs (5-FU, capecitabine) and resulting in chemoresistance [37]. PI3K/mTOR inhibition in CRC patient-derived cells has also been reported to reinstate cetuximab resistance [45]. PI3K action is regulated by the tumor suppressor phosphatase and tensin homolog (PTEN). Altered pathways responding to insulin, estrogen, and androgen involve PTEN [46], resulting in cetuximab and panitumumab resistance [47]. Thus, PI3K/Akt/mTOR signaling is effectively involved in many hormonal pathway-related resistances, though it is not the only one. Upon binding of ligands (such as insulin and growth factors) to their tyrosine kinase receptor [48], PTEN has also been involved in cetuximab resistance by altering Ras pathway activity [49]. MAPK/ERK pathway was found downstream of many growth factor receptors often overexpressed in CRC [50], such as insulin-like growth factor receptor (IGFR) [51]. MAPKs are involved in several cellular functions such as proliferation, differentiation, and apoptosis through three primary components: p38, ERK, and JNK. Hyperactivation of p38 was recently implicated in oxaliplatin resistance in CRC [52]. MAPKs exert their functions through NF-kB [46], activated in 60–80% of CRC patients [53], and involved in conferring chemoresistance through inhibition of apoptosis [54]. In other studies, CRC cells resistant to 5-FU showed high expression of TCF4 and β-catenin, indicating an altered Wnt pathway [55]. Since hormone signaling is triggered upon binding of the hormone to its receptor, expression of hormone receptors in recipient cells is important in modulating the impact of this interaction, as well as influencing downstream cascades. The following subsections describe different hormonal systems altered in CRC, which may lead to chemoresistance.

### 2.1. Insulin/Insulin-Like Growth Factor Signaling Pathway in CRC

The insulin/IGF system plays a crucial role in CRC development [56]. Insulin/IGF bind to the insulin receptor (IR), which is converted into its activated phosphorylated form. High levels of pIR were observed in the transition phase from normal epithelium to adenocarcinoma, indicating the key role of insulin/IR in CRC progression [57]. IR substrates (IRS)-1 and -2 also contribute to disease development: IRS-1 overexpression is associated with metastasis [58], whereas IRS-2 is related to early-stage tumor formation. IRS-2 overexpression increases PI3K pathway activity, leading to Akt phosphorylation and reduced cell adhesion in colon cancer [59]. IGF-I and IGF-II are significantly overexpressed in CRC [60]. High IGF serum levels were detected in CRC patients and found to act through autocrine/paracrine as well as endocrine signaling [51]. IGF regulates cell survival, proliferation, and metabolism by binding to IGFR or IR. IGF-1R was found overexpressed in CRC [61] where it resulted in Akt activation and upregulation of the anti-apoptotic protein Bcl-xL [62]. IGF-1R is associated with drug resistance as it regulates the expression of multidrug-resistance-associated protein 2 (MRP2). For this purpose, Shen et al. silenced IGF-1R which subsequently drove the nuclear translocation of nuclear-factor-like2 and blockage of *MRP2* gene expression [63]. Low *MRP2* expression causes an increase in intracellular drug concentration and accumulation, suggesting how the IGF-1R signaling pathway may foster drug efflux and tumor cell survival despite therapy. Further analysis of these hormone receptors revealed that silencing of IGF-1R increases the sensitivity of CRC cells to 5 -FU, mitomycin C, oxaliplatin, and vincristine [63].

### 2.2. Thyroid Hormone Signaling Pathway in CRC

Animal studies and clinical data from CRC patients suggest that thyroid status may impact tumor formation, growth, metastasis, and drug sensitivity [64,65,66]. This physiological hormonal system comprises both triiodothyronine (T3) and thyroxine (T4) hormones. T4 is the primary inactive thyroid product, which is converted into active T3 by type 1 or 2 deiodinase enzymes (D1 and D2, respectively). In contrast, type 3 deiodinase (D3) transforms T4 and T3 into inactive metabolites [65,67]. Deiodinases represent the link between thyroid hormone signaling and β-catenin pathway. The β-catenin/TCF complex regulates D3 expression at the transcriptional level resulting in high expression of D3 in CRC. Indeed, using E-cadherin to sequester β-catenin, a reduction in D3 expression occurs.Low D3 levels lead to increased expression of T3, resulting in inhibited proliferation of CRC cells [68]. Furthermore, T3 may induce resistance to oxaliplatin, 5-FU, and their combination (FOX) by downregulating Akt/PI3K in CRC stem cells [69]. Basically, T3 binds to its receptors (TRα and TRβ), regulates the expression of target genes via thyroid hormone receptor response elements [70]. Thyroid hormone receptor (TR) β1 expression is downregulated in CRC, prompting cancer cells to proliferate and migrate via PI3K/Akt signaling modulation. Overexpression of TRβ1 induces a reduction of inactivated Akt, disrupting its pathway and suppressing CRC cell progression and migration [71]. In contrast, TRα1 overexpression and Wnt signaling pathway upregulation promote tumor formation in CRC [72] by contributing to escape from chemotherapy-induced apoptosis [73]. Indeed, further evidence report that T3 affects drug resistance targeting BMP4/Wnt pathway [70].

### 2.3. Additional Altered Hormone Signaling in Resistant CRC Cells

Other hormonal systems contribute to drug resistance in CRC cells. Leptin is an adipokine involved in regulating hyperphagia, glucose homeostasis, growth, immune response, and angiogenesis [74]. Leptin and its receptors (ObR) are highly expressed in CRC [75], indeed, leptin inhibition reduces tumor growth [76] and leads to an overexpression of ObR, which inhibits 5-FU-induced apoptosis [77]. Moreover, high levels of growth hormone (GH) were identified in CRC patients, where the hormone enhances cell proliferation, survival, and oncogenicity through autocrine signaling [78]. GH also plays a role in the acquisition of drug-induced apoptosis resistance by inhibiting the expression of PPARγ and BAX in CRC cells, allowing evasion from therapy-mediated DNA damage [79]. Corticotropin-releasing hormone (CRH) is a peptide hormone involved in stress response that exerts its function by binding to its specific receptors, including corticotropin-releasing hormone receptor2 (CRHR2), which exhibits a significantly reduced expression in CRC. Fas is one of the CRHR2/Ucn2 signaling targets in CRC and alternation in Fas expression prevents Fas-mediated apoptosis, thus conferring resistance to therapy [80]. The lower incidence of CRC in women than in men highlights the importance of estrogen as a biological protective effector in gastrointestinal disease [81,82]. Estrogens can bind to estrogen receptor α (ERα) orβ (ERβ). In normal colon regions, ERβ is implicated in epithelium maintenance and growth, and in immune system modulation, while decreased ERβ expression promotes the risk of CRC partly by influencing gut permeability [83]. Reduced ERβ expression in CRC was also correlated with increased proliferation and inhibition of apoptosis [84]. ERα downregulation in CRC [85] suggests that this receptor might contribute to drug resistance via regulatory effect in the expression of pro- or anti-apoptotic proteins, such as cyclin D1 [84]. Estrogen signaling might be indirectly involved in conferring resistance in CRC cells since a deficiency in ER signaling can be compensated for by other signaling pathways in these cells [86,87]. Estrogen-related receptor alpha (ERRα) is also involved in the same system. ERRα is an orphan nuclear receptor since its endogenous ligand has not yet been identified. Interestingly, it is constitutively active even without any ligand binding [88] and regulates the expression of many genes, most of which are also targeted by ERα [89]. ERRα is overexpressed in CRC [83] and its upregulation is responsible for resistance to trametinib [90], making ERRα a potential new target in CRC.

## 3. Role of miRNA in Hormone Signaling in Therapy-Resistant CRC

miRNAs are known as posttranscriptional regulators of gene expression, affecting about 60% of overall protein-coding genes [91], thereby modulating developmental, physiological, and pathological processes. Recent reports described their critical role in the regulation of hormone signaling, including expression modulation of hormone receptor-coding genes and genes related to hormonal intracellular signaling cascades [92,93,94]. Moreover, miRNAs can be modulated by hormones [95], thus suggesting a reciprocal flux which anyway revises cancer cell sensitivity to treatment. Several deregulated miRNAs are associated with drug resistance in CRC (Table 1), many of which are directly involved in hormone signaling. The following sections unfold the involvement of miRNAs into hormonal signaling pathways whose interactions are responsible for decreased therapy success by overcoming drug-mediated apoptosis and, thereby, inducing survival in CRC cells.

### 3.1. miRNAs and Insulin-Like Growth Factor (IGF) Receptor

IGFs are implicated in tumorigenesis of different cancers, such as prostate, breast, and CRC [51]. As explained in sub-paragraph 2, both IGFs and IGF-R were found overexpressed in CRC, and such boosted IGF-I/IGF-IR signaling pathway enhances cell survival and resistance to chemotherapy [62], a process that apparently involves CIMP. The chromosomal segment 17p13.1 is frequently deleted in CRC, resulting in reduced expression of both miR-497 and miR-195, as their encoding sequences are located in this chromosomal region [96]. Guo et al. proposed miR-497 to have a role in inhibiting IGF-IR signaling pathway, indeed, in CRC, low miR-497 is associated with higher IGF-IR levels, which, in turn, modulates cell death. Restoration of miR-497 expression could be useful for inhibition of IGF-IR in CRC and resistance to apoptosis by blocking overactivation of survival signaling pathways [96]. Many other miRNAs are implicated in IGFR signaling modulation resulting in drug resistance. As a matter of fact, miR-143 targets IGF-1R resulting in oxaliplatin resistance [97] and, indeed, overexpression of miR-143 was reported to re-sensitize CRC cells to such therapeutic by caspases activation [97]. Interestingly, Xu et al. found that miR-143 is downregulated in blood as well, suggesting its potential as a prognostic biomarker in CRC. Besides miR-143, IGF-1R is also a target of miR-302, which downregulation in CRC may cause 5-FU resistance. Liu et al., actually found that miR-302 overexpression reduces 5-FU resistance as demonstrated by enhanced cell death, subsequently, therapeutic treatment. This re-sensitizing action is due to the direct repression of IGF-1R associated with consequent inhibition of Akt, frequently activated in chemo-resistant CRC cells [98]. Lastly, a further miRNA is able to modulate IGFRs expression skewing CRC therapeutic response. In this respect, miR-185 was found to be involved in radio-resistance in CRC cells by targeting IGF-1R and IGF-2R. In detail, miR-185 is downregulated in colon cancer, leading to increased expression of its targets and, thus, resulting in refractory to ionizing radiation, as it was confirmed by improved CRC cell radio-sensitivity upon restoration of the miRNA expression [99].

### 3.2. miRNAs and IGF-R Downstream Components

Besides receptors, miRNAs impair hormone signaling also at the downstream level, resulting in ectopic pathways and then in drug resistance. Among such regulators, miR-1260b targets PDCD4, one of the effectors of the PI3K/Akt signaling pathway reported to regulate apoptosis [100] and the response to drug treatment [101]. miR-1260b expression was reported increased in CRC patients and linked to 5-FU therapy resistance. Accordingly, the use of a miR1260b inhibitor confirmed the involvement of such miRNA in 5-FU-resistance in CRC, further revealing to be mediated by inhibition of PI3K/Akt signaling pathway. Interestingly, the overexpression of IGF-I in CRC enhanced the activation of such pathway, indicating that miR-1260b may regulate drug sensitivity via IGF signal modulation [102]. IRS-1 is reported to be an oncogene involved in the regulation of angiogenesis, metastasis, and even chemosensitivity, thereby, IRS-1 increased levels due to the reduction of its regulatory miRNA expression may be responsible for lowering the therapy responsiveness in CRC. In the same manner, miR-145 exhibits a tumor suppressor activity in CRC by modulating Myc [103], STAT1 [104] and other genes including IRS-1 [105].In fact, miR-145 downregulation was shown to increase tumor survival [106], a feature sharply in contrast with the therapeutic success. Lastly amidst other miRNAs, miR-128 is downregulated in CRC, failing to restrain IRS-1 and resulting in inhibited apoptosis [107].

### 3.3. miR-155 and Adrenaline

Previous examples have exhibited a definite trend, where miRNAs were responsible for the refractory phenotype in CRC by skewing hormone signaling pathways. Nevertheless, there is evidence that a different course exists since hormones may affect miRNAs expression. The miR-155/adrenaline coupling is one example, as adrenaline was found to increase miR-155 expression, leading to drug resistance [108]. CRC cells significantly express adrenergic receptors [109] whose ligands are catecholamines, including adrenaline. Adrenaline was previously shown to induce 5-FU resistance in CRC in vitro by upregulating ABCB1 transporter [109].Indeed, mouse cancer models exposed to stressful stimuli to set up catecholamine signals, exhibited restrained response to anti-tumor therapy [110]. In this regard, adrenaline showed to activate NF-kB signaling in CRC cells, increasing the expression of miR-155, which, in turn, favored cell proliferation and reduced sensitivity to cisplatin [110], corroborating the potential of catecholamines in endowing resistance to CRC.

### 3.4. miR-7 and Corticotropin-Releasing Hormone Receptor

Nowadays, immune therapy has emerged as a rising approach to the conventional anti-tumor treatments since cancer cells were shown to express a large amount of inhibitory molecules, such as PD-L1 [111], CD80/CD86 [112], which undermine the immunological response through binding immune checkpoint receptors such as PD-1 and CTLA4 [113]. Another way exploited by cancer cells to deceive the immune-mediated cytotoxicity is the escape from Fas/FasL-mediated apoptosis [114,115,116]. Although cancer treatments, such as radiotherapy, exhibited a successful increase in Fas expression [115], immune escape is still a current challenge and the CRH system seems to be involved in it. Rodriguez et al. reported that CRHR2 expression is reduced in CRC cells supporting tumor survival, proliferation, EMT, metastasis, and resistance [117]. In vitro and in vivo studies showed that miR-7 expression is induced by CRHR2 signaling and its ectopic expression enhances apoptosis and cell cycle arrested by negatively regulating YY1 expression [118]. Thus, YY1 expression benefits from reduced CRHR signaling and its overexpression impairs Fas expression. Collectively, CRHR2/Ucn2 signaling sustains CRC cell resistance to Fas/FasL-mediated apoptosis by targeting miR-7/YY1/Fas circuit [80], as it was confirmed by increased CD-11 (Fas agonist)-mediated apoptosis after restoration of CRHR2 signaling.

## 4. Novel Therapeutic Agents in the Treatment of Drug Resistance in CRC

Many therapeutics acting on hormone signaling have been tested towards resistant CRC. In CRC, neurokinin receptor 1 (NK1R) antagonists reduced tumor growth through inhibition of Wnt/β-catenin and Akt/mTOR signaling pathways. The NKIR antagonist also inhibited cancer stem cells, which are thought to confer higher resistance to therapy [119]. Besides molecule antagonists, hormone analogs have been tested in CRC. Lee et al. exploited a T4 analog (named Tetrac) to block CRC progression competing with the endogenous hormone to bind and, thus, block the surface integrin receptor αvβ3. Its usage, in combination with cetuximab, was reported to inhibit cell proliferation and CRC xenografts [120]. Brigatinib, which inhibits IGF-1R [121], and Amiodarone, a Thyroid Hormone Receptor antagonist [122], were studied with encouraging results. Significant outcomes have also been obtained with natural products in resistant CRC cells. Gambogic acid is a natural compound extracted from the *Garciniahanburyi* plant [123]. It acts by interfering with MAPK signaling via JNK pathway, which, in turn, induces apoptosis and inhibits proliferation of 5-FU-resistant CRC cells [123]. The China Food and Drug Administration has approved its use in phase II clinical trials for the treatment of solid tumors. Piperlongumine, a natural constituent of the fruit of the long pepper (Piper longum), inhibits PI3K and Ras, reducing the activity of Akt/NF-kB, c-Myc, cyclin D1, blocking CRC cell growth, proliferation and survival. It also induces mitochondrial apoptosis, downregulating Bcl-2 [124]. In addition to the molecules that directly modulate pathway component activity and considering the emerging role of miRNAs in regulating the expression of genes involved in hormone signaling pathways, recent studies have been focused on strategies to regulate oncomiR expression, including antisense anti-miR oligonucleotides, locked nucleic acid anti-miRs, miRNA sponges and small molecule inhibitors of miRNAs (SMIRs) [125]. ASMIR against miR-21, named AC1MMYR2, was developed by Shi et al., who studied the three-dimensional structure of the Dicer binding site on pre-miR-21 to find a molecule able to block miR-21 maturation. AC1MMYR2 inhibits the expression of miR-21 and consequently increases the expression of its targets: PTEN (Figure 2), PDCD4 and RECK [126]. This axis is involved in 5-FU resistance (Table 1). miR-1260b inhibitor was tested by Zhao et al., who showed the reduction of PDCD4 expression and p-Ant and p-PI3K levels in IGF-R signaling pathway, thus, re-sensitizing CRC to 5-FU treatment [102]. One of the most used anti-miRs in CRC is anti-miR-135b [127,128]. Upregulation of miR-135b in CRC reduces apoptosis and increases cell growth by regulating the expression of TGF-βR2, DAPK1, APC and FIH by activating APC/β-catenin and Src-PI3K pathways. Anti-miR-135b was therefore used in a CRC mouse model, resulting in reduced proliferation and increased apoptosis [129] and drug sensitivity [128]. SD-208, a TGF-β receptor kinase inhibitor, downregulated miR-135b expression [127]. Among other targets, miR-135b affects FOXO1 and ERRα expression, which is associated with apoptosis modulation (Figure 2). FOXO1 is also involved in IR signaling cascade [130]. Thus, pre-treatment with SD-208 improved chemosensitivity in CRC cells resistant to 5-FU [127]. Melatonin was shown to exert a robust anti-tumor effect through modulation of cell cycle dynamics and apoptosis [131]. Recently, Sakatani et al. investigated the molecular mechanism of melatonin in 5-FU-resistant CRC cells [132], suggesting that melatonin strongly promotes apoptosis in 5-FU-resistant CRC cells by blocking thymidylate synthase (TYMS) activity, pointing to an inverse correlation between TYMS expression levels and 5-FU sensitivity in CRC. TYMS is one of the main downstream targets of miR-215-5p. Melatonin upregulates miR-125-5p expression, enhancing the suppressive effect of miR-125-5p on TYMS expression in 5-FU-resistant cells, thus, leading to inhibition of proliferation and increased apoptosis in these resistant cells [132].

## 5. Conclusions

The current knowledge of molecular mechanisms of resistance to cytotoxic agents and targeted therapies for CRC treatment indicates the existence of complex genetic, epigenetic, and metabolic alterations leading to impaired drug sensitivity and multiple mechanisms for drug resistance. 5-FU remains the first-line therapy in CRC, but new strategies have been pursued to improve survival and therapeutic response rates, mainly in metastatic or advanced stages of CRC. Drugs, such as oxaliplatin, irinotecan, and capecitabine were first developed and followed by combination therapies or pre-treatments. The development of novel and more effective treatments is an urgent need. The involvement of aberrant hormone signaling in conferring resistance to many therapeutic approaches in CRC has been reported. miRNAs seem to play a prominent role in this scenario. Several miRNAs are aberrantly expressed in CRC, resulting in altered hormonal signaling and therapy resistance, including miR-7 in the CRHR2 system, which leads to immune resistance; miR-302, miR-497, miR-185 and miR-143 in the IGF system, which lead to 5-FU and oxaliplatin resistance, respectively; miR-125, which regulates expression of TYMS resulting in 5-FU resistance and miR-1260b, which regulates PDCD4 expression leading to 5-FU resistance. Although the role of miRNAs in hormone signaling and drug resistance/sensitivity in CRC needs to be further investigated, the preliminary evidence is an encouraging step forward to identify novel therapeutic approaches against acquired drug resistance and useful for patients stratification.

## Figures and Tables

**Figure 1 cells-10-00039-f001:**
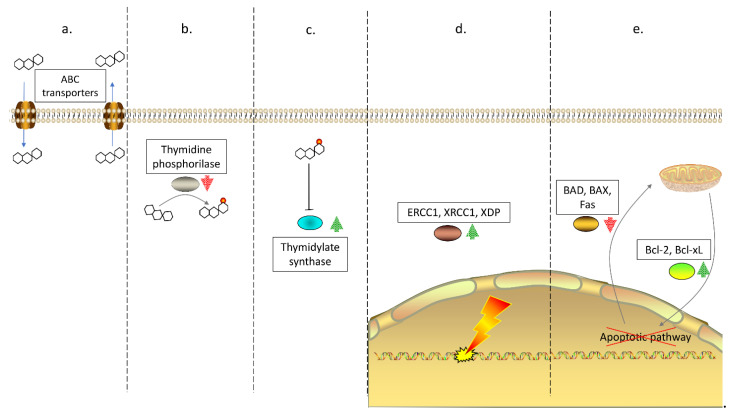
Drug resistance mechanisms in CRC. (**a**). Unbalance between drug uptake and efflux, reducing cellular drug concentration and thus treatment efficacy; (**b**). Prevention of drug activation: when a drug-activating enzyme such as Thymidine Phosphorylase (TP) for 5-FU and capecitabine is lacking or altered, the drug remains in an inactive form preventing its function; (**c**). Aberrant target molecule expression: drugs act on specific target molecules, but whether the target is overexpressed, such as Thymidylate Synthase (TS) for 5-FU treatment, the therapy fails; (**d**). Self-renewal: resistant cancer cells increase expression of proteins involved in DNA repair such as Excision Repair Cross Complementing 1 (ERCC1) and X-Ray Repair Cross Complementing 1 (XRCC1), rendering radiation therapy ineffective, as radiotherapy-induced DNA damage is then repaired by proteins; (**e**). Apoptosis escape: many drugs work by activating apoptotic pathways in cancer cells; however, resistant CRC cells increase expression of anti-apoptotic proteins (Bcl-2, Bcl-xL) and decrease expression of pro-apoptotic proteins (BAD, BAX, and Fas) thus promoting cell survival.

**Figure 2 cells-10-00039-f002:**
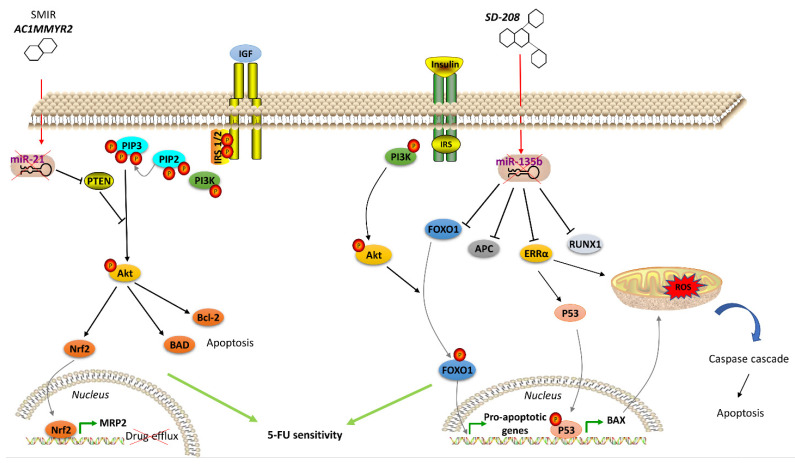
Agents acting on hormone signaling to revert drug resistance in CRC. Left, the small molecular inhibitor of specific miRNA (SMIR) AC1MMYR2 represses miR-21, which regulates PTEN expression, thus modulating IGFR signaling pathway and resulting in inhibited drug efflux and induced apoptosis in resistant CRC cells. Right, SD-208 inhibits miR-135b, which controls expression of proteins (including FOXO1 in IR signaling pathway) involved in apoptosis regulation, thus resulting in apoptosis of resistant CRC cells. (↑ = miRNA overexpression) (↓ = miRNA downexpression)

**Table 1 cells-10-00039-t001:** Drug resistance-associated miRNAs in CRC.

miRNA	Expression	Targets	Resistance	Reference
miR-21	↑	PDCD4 TIAM1 PTEN TGFBR2 CDC25A	5-FU	[120]
miR-143	↓	NF-kB Bcl-2; IGF-1R	5-FU	[94,121]
miR-365; miR-224	↓	-	5-FU	[122]
miR-34a	↓	SIRT1	5-FU	[123]
miR-494	↓	DPYD	5-FU	[124]
miR-302	↓	IGF-1R	5-FU	[133]
miR-10b	↑	BIM	5-FU	[134]
miR-375-3p	↓	TYMS	5-FU	[135]
miR-20a	↑	BNIP2	5-FU OxaliplatinTeniposide	[136]
miR-320	↓	FOXM1	5-FU Oxaliplatin	[127]
miR-34a	↑	-	Oxaliplatin	[129]
let-7g	↓	Cyclin D Myc E2F	S-1	[125]

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
