# Peer review of "MicroRNA-Assisted Hormone Cell Signaling in Colorectal Cancer Resistance"

_cells, 2020, doi:10.3390/cells10010039_

Round 1
Reviewer 1 Report
The manuscript is important to the field and presents good review of studies. I would recommend to do Figures with better larger font labels of signaling molecules. I would add more clinical trials to Discussion with drugs, pathways presented.
Author Response
We kindly thank you for your careful revision.
According with your suggestion about Figures, we modified Graphical abstract, Figure 1 and 2 improving the size of font and their resolution. In addition, we appreciated your suggestion in discussion part but although there are clinical trials regarding the use of therapeutic drugs against miRNAs or hormonal targets in CRC, such as JMR-132 (24876732) and Regorafenib (28236743) respectively, we preferred to report only clinical trials involving combined hormone-miRNAs action.
Reviewer 2 Report
The review ‘MicroRNA-assisted hormone cell signaling in colorectal cancer resistance’ summarizes the role of microRNA in hormone signaling pathways associated with colorectal cancer drug resistance. Overall, the review provides an extensive review of the current literature, yet minor revisions need to be completed.The specific points that resulted in this conclusion are listed below.
Line 38: Add ‘in’ between ‘(CIMP)’ and ‘~10-20%’.
Throughout the manuscript, there are several words that run together. This may have occurred during formatting. Check entire manuscript. Examples include Line 63-64 with ‘hasbeen’ and ‘mostlytocisplatain’.
Line 225: Change ‘alfa’ to ‘alpha’.
Line 238: Add comma after ‘(Table 1)’.
While the authors have referenced over 130 papers for this review, there are several paragraphs that do not read well. The paragraphs just include a collection of referenced results of the papers. There is little discussion of tying everything together. For example, in Lines 279-280, the last sentence refers to the abundance of miR-128. However, this is the only sentence discussing miR-128 and it is not tied into the discussion of the other miR that are discussed in the paragraph. In fact, that is the only reference to miR-128 in the review. The entire manuscript needs to be edited for this.
The figures are a helpful addition to show the regulatory pathways that may be affected.

Author Response
We kindly thank you for detailed reading of our work. We are sorry for formatting problem. We revised all the manuscripts and corrected the spelling errors and spaces lacking. Below, we listed them.
English and grammar edition:
Line 28: applied space before word “microRNA” in keywords.
Line 68: Added comma after word “indeed”
Line 127: Space correction in the word “mediated by”
Line 133: Added space between dot and the word “Altered”
Line 135: Added comma after the word “Thus”
Line 148: Added “s” after word “cascade”
Line 151: Deleted comma after “IGF system”.
Line 161: Added space before the word “where”
Line 181: Added space between “downregulating” and “Akt/PI3K”
Line 191: Added number “2.3” before subtitle
Line 211: Added space between “Cyclin” and “D1”
Line 232: Change the capital “I” into small “i” in the word “insulin-like growth factor” and also added “IGF” after it.
Line 262: Added space between “3.2” and “miRNA” in subtitle
Line 271: deleted space between “down” and “regulated” in the word “downregulated”
Line 272: Added comma after the word “indeed”
Line 277: Changed the word “de facto” into the word “in fact”. Deleted space between “down” and “regulated” in the word “downregulated”
Line 280: Deleted space between “3.3” and “miR-155”
Line 287: Added comma after word “indeed”
Line 294: Deleted space between “3.4”and “miR-7” in subtitle
Line 295: Added comma after the word “Nowadays”
Line 300: Deleted comma after the word “treatment”
Line 305: Added “-ed” after the word “arrest”
Line 318: Change “use” to “usage”
Line 327: Deleted two extra “and” before “Cyclin D1” and “blocking”. Deleted comma after “proliferation”
Line 328: Added “the” before the word “molecule”
Line 330: Added “been” between “have” and “focused”
Line 333: Deleted comma before “and”
Line 333: Put “named AC1MMYR2” between two comma
Line 335: Added “the” before “expression”
Line 336: Deleted comma after PDCD4
Line 338: Added comma after “thus”
Line 341: Deleted comma after “APC” and deleted “and” after “FIH”
Line 361: Deleted comma after “CRC”
Line 362: Change comma to “and” between the words “developed” and “followed”
Line 372: Added space between “miRNAs” and “are”
Moreover, according your consideration, we edited some point of discussion in subparagraphs 3.1, 3.2, 3.3 and 3.4. Paragraph 3 was overall modified in order to tie better together different parts/findings. The applied modifications are marked in yellow.
Reviewer 3 Report
In their manuscript entitled “MicroRNA-assisted hormone cell signaling in colorectal cancer resistance”, Massaro and colleagues review the role of miRNAs in hormone signaling pathways in CRC drug resistances. This review is very informative and covers the recent literature, the conclusion is adequate, however, following comments on Figures and Table1 need to be improved by the authors:
- It is easier for the reader to understand Table 1 by grouping “Upregulated” or “Downregulated” in the “Expression” column or the type of drug in the “Resistance” column.
- The letters are too small in all figures, so please make them larger overall including the letters.
Author Response
We appreciate the insightful your suggestions and we have proceeded to modify the table 1 by grouping of the type of drug in the “Resistance” column. Moreover, Table 1 was modified in the column “Expression” adding arrows instead of the written explanation. We thank you because in this way the table is more easier for the reader understanding.
In the same time, According with your suggestion about Figures, we modified Graphical abstract, Figure 1 and 2 improving the size of font and their resolution.